# Diffeomorphic Explanations with Normalizing Flows

**Ann-Kathrin Dombrowski** [* 1]    **Jan E. Gerken** [* 2]    **Pan Kessel** [1 3]

## Abstract

Normalizing flows are diffeomorphisms which are parameterized by neural networks. As a result, they can induce coordinate transformations in the tangent space of the data manifold. In this work, we demonstrate that such transformations can be used to generate interpretable explanations for decisions of neural networks. More specifically, we perform gradient ascent in the base space of the flow to generate counterfactuals which are classified with great confidence as a specified target class. We analyze this generation process theoretically using Riemannian differential geometry and establish a rigorous theoretical connection between gradient ascent on the data manifold and in the base space of the flow.

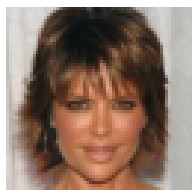 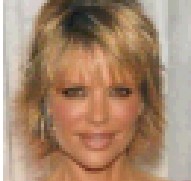 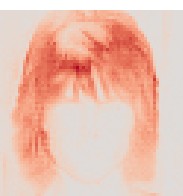

original $x$           counterfactual $x'$          heatmap $\delta x$

*Figure 1.* Diffeomorphic explanation for hair-color classification.

## 1. Introduction

Explaining a complex system can be drastically simplified using a suitable coordinate system. As an example, the solar system can be explained either by using a reference system for which the sun is at rest (heliocentristic) or, alternatively, for which the earth is at rest (geocentristic). Despite

wide-held belief, both reference system are physically valid. However, the dynamics of the planets is significantly easier to describe in heliocentristic coordinates since the planets will follow geometrically simple trajectories.

Explanation methods for neural networks have recently gained significant attention because they promise to make black-box classifiers more transparent, see (Samek et al., 2019) for a detailed overview. In this paper, we use the bijectivity of a normalizing flow to consider a classifier in the base space of the flow. This amounts to a coordinate transformation in the data space (or mathematically more precise: a diffeomorphism). We will show that in this coordinate system, the classifier is more easily interpretable and can be used to construct counterfactual explanations that lie on the data manifold. Using Riemannian differential geometry, we will analyze the advantages of creating counterfactual explanations in the base space of the flow and establish a process by which the tangent space of the data manifold can be estimated from the flow. We strongly expect these theoretical insights to be useful beyond explainability.

In summary, our main contributions are as follows:

- We propose a novel application domain for flows: inducing a bijective transformation to a more interpretable space on which counterfactuals can be easily generated.
- We analyze the properties of this generation process theoretically using Riemannian differential geometry.
- We experimentally demonstrate superior performance compared to more traditional approaches for generating counterfactuals for classification tasks in three different domains.

## 2. Counterfactual Explanations

Let $f : X \to \mathbb{R}^K$ be a classifier whose component $f(x)_k$ is the probability for the point $x \in X$ to be of class $k \in \{1, \dots, K\}$. We make no assumptions on the architecture of the classifier $f$ and only require that we can evaluate $f(x)$ and its derivative $\partial_x f(x)$ for a given input $x \in X$.[1]

In this work, we will follow the well-established paradigm of *counterfactual explanations* – see (Verma et al., 2020) for

---

[*]Equal contribution [1]Machine Learning Group, Department of Electrical Engineering & Computer Science, Technische Universität Berlin, Germany [2]Department of Mathematical Sciences, Chalmers University of Technology, Gothenburg, Sweden [3]BIFOLD - Berlin Institute for the Foundations of Learning and Data, Technische Universität Berlin, Berlin, Germany. Correspondence to: Pan Kessel <pan.kessel@tu-berlin.de>.

Third workshop on *Invertible Neural Networks, Normalizing Flows, and Explicit Likelihood Models* (ICML 2021). Copyright 2021 by the author(s).

---

[1]This assumption can be relaxed: if we do not have access to the gradient, we can approximate it by finite differences.

a recent review. These methods aim to explain the classifier $f$ by providing an answer to the question which minimal deformations $x' = x + \delta x$ need to be applied to the original input $x$ in order to change its prediction. Often, the difference $\delta x$ is then visualized by a heatmap highlighting the relevant pixels for the change in classification, see Figure 1 for an example.

In the following, we will assume that the data lies on a submanifold $S \subset X$ which is of (significantly) lower dimension $n$ than the dimension $N$ of its embedding space $X$. We stress that this is also known as the manifold assumption and is expected to hold across a wide range of machine learning tasks, see e.g. (Goodfellow et al., 2016). In these situations, we are often interested in only the deformations $x'$ which lie on the data manifold $S$. As an example, a customer of a bank may want to understand how their financial data needs to change in order to receive a loan. If the classification changes off-manifold, for example for zip codes that do not exist, this is of no relevance since the user is obliged to enter their correct zip code. Furthermore, it is often required that the deformation is minimal, i.e. the perturbation $\delta x$ should be as small as possible. However, the relevant norm is the one of the data manifold $S$ and not of its embedding pixel space $X$. For example, a slightly rotated number in an MNIST image may have large pixel-wise distance but should be considered an infinitesimal perturbation of the original image.

More precisely, we define counterfactuals as follows: let $t = \operatorname{argmax}_j f_j(x)$ be the predicted class for the data point $x \in S$. The set $\Delta_{k,\delta} \subset S$ of *counterfactuals* $x'$ of the point $x$ with respect to the target class $k \in \{1, \ldots, K\} \setminus \{t\}$ and confidence $\delta \in (0, 1]$ is defined by

$$\Delta_{k,\delta} = \{x'(x) \in S : \quad \operatorname{argmax}_j f_j(x') = k \wedge f_k(x') > \delta\},$$

i.e. all points on the data manifold which are classified to be of the target class $k$ with at least the confidence $\delta$. A *minimal counterfactual* $x' \in \Delta_{k,\delta}$ is then a counterfactual with smallest distance $d_\gamma(x', x)$ to the original point $x$, where $d_\gamma$ is the distance on the data manifold (induced by its Riemannian metric $\gamma$). Note that there may not be a unique minimal counterfactual.

## 3. Construction of Counterfactual

We propose to estimate the minimal counterfactual $x'$ of the data sample $x$ with respect to the classifier $f$ by using a diffeomorphism $g : Z \to X$ modelled by a normalizing flow.

The flow $g$ equips the space $X$ with a probability density

$$q_X(x) = q_Z(g^{-1}(x)) \left| \det \frac{\partial z}{\partial x} \right| \tag{1}$$

by push-forward of a simple base density $q_Z$, such as $N(0,1)$, defined on the base space $Z$. We assume that the flow was successfully trained to approximate the data distribution $p_X$ by minimizing the forward KL-divergence as usual (this assumption will be made more precise in Section 4).

We then perform gradient ascent in the base space $Z$ to maximize the probability of the target class $k$, i.e.

$$z^{(t+1)} = z^{(t)} + \lambda \frac{\partial (f \circ g)_k}{\partial z}(z^{(t)}), \tag{2}$$

where $\lambda$ is the learning rate and we initialize by mapping the original point $x$ to the base space by $z^{(0)} = g^{-1}(x)$. We then take the sample $x^{(T)} = g(z^{(T)})$ as an estimator for a minimal counterfactual if $x^{(T)}$ is the first optimization step to be classified as the target $k$ with given confidence $\delta$:

$$\operatorname{argmax}_j f_j(x^{(T)}) = k \quad \text{and} \quad f_k(x^{(T)}) > \delta \,.$$

This is because generically taking further steps only increases the distance to the original sample $||z^{(T)} - z^0|| > ||z^{T+t} - z^0||$ for $t > 0$ and we want to find (an estimate of a) minimal counterfactual. This may also be validated by continuing optimization for a certain number of steps and selecting the sample with the minimal distance.

As discussed in Section 6, generative-model-based methods to estimate (minimal) counterfactuals have previously been proposed, for example based on Generative Adversarial Networks or Autoencoders. However, the relevance of normalizing flows in this domain has so far not be recognized. This is unfortunate as normalizing flows have important advantages in this application domain compared to other generative models: firstly, a flow $g$ is a diffeomorphism and therefore no information is lost by considering the classifier $f \circ g$ on $Z$ instead of the original classifier $f$ on $X$, i.e. there is a unique $z = g^{-1}(x) \in Z$ for any data point $x \in X$. Secondly, performing gradient ascent in the base space $Z$ of a well-trained flow will ensure (to good approximation) that each optimization step $x^{(t)} = g(z^{(t)})$ will stay on the data manifold $S \subset X$. Since the base space $Z$ has the same dimension as the data space $X$, the latter statement is far from obvious and is substantiated with theoretical arguments in the next section.

## 4. Theoretical Analysis

In the following, it will be shown that performing gradient ascent in $Z$ space and then mapping the result in $X$ space will stay on the data manifold $S$.

This is in stark contrast to gradient ascent directly in $X$ space, i.e.

$$x^{(t+1)} = x^{(t)} + \lambda \frac{\partial f_k}{\partial x}(x^{(t)}), \tag{3}$$

where $\lambda$ is the learning rate. It is well-known that such an optimization procedure would very quickly leave the data manifold $S$, see for example (Goodfellow et al., 2015).

For gradient ascent in $Z$ space (2), each step $z^{(t)}$ can uniquely be mapped to $X$ space by $x^{(t)} = g(z^{(t)})$. In the Supplement A.1, we derive the following result:

**Theorem 1.** *Let $z^{(t)}$ be defined as in (2) and $x^{(t)} = g(z^{(t)})$. Then, to leading order in the learning rate $\lambda$,*

$$x^{(t+1)} = x^{(t)} + \lambda \, \gamma^{-1}|_{g^{-1}(x^{(t)})} \frac{\partial f_k}{\partial x}(x^{(t)}) + \mathcal{O}(\lambda^2) , \quad (4)$$

*where $\gamma^{-1} = \frac{\partial g}{\partial z} \frac{\partial g}{\partial z}^T \in \mathbb{R}^{N,N}$ is the pull-back of the flat metric on $Z$ under the flow $g$.*

Therefore, performing gradient ascent in $X$ or $Z$ space is not equivalent because (3) and (4) do not agree. In particular, the presence of the inverse metric $\gamma^{-1}$ in the update formula (4) effectively induces separate learning rates for each direction in tangent space.

In the following, we prove that directions orthogonal to the data manifold $S \subset X$ are heavily suppressed by the inverse metric and thus gradient ascent (4) stays on the data manifold $S$ to very good approximation.

In practice, the data manifold is only *approximately* of lower dimension. Specifically, we assume that the data manifold is a product manifold, equipped with the canonical product metric, of the form

$$S = \mathcal{D} \times B_{\delta_1} \times \cdots \times B_{\delta_{N-n}} , \quad (5)$$

where $\mathcal{D}$ is a $n$-dimensional manifold and $B_\delta$ is an open one-dimensional ball with radius $\delta$ (with respect to the flat metric of the embedding space $X$). Since we will choose all the radii $\delta_i$ to be small, the data manifold $S$ is thus approximately $n$-dimensional.

We choose Gaussian normal coordinates $x = (x_\parallel^1, \ldots, x_\parallel^n, x_\perp^1, \ldots, x_\perp^{N-n})$ on $X$, where the $x_\perp^i$ are slice coordinates for $B_{\delta_i}$ and the $(x_\parallel^1, \ldots, x_\parallel^n)$ are slice coordinates of $\mathcal{D}$, see Figure 2. We furthermore require that in our coordinates $x_\perp^i(p) \in (-\delta, +\delta)$ for $p \in S$. We then show in the Supplement A.2:

**Theorem 2.** *Let $p_X$ denote the data density with $\mathrm{supp}(p_X) = S$, and the flow $g$ be well-trained such that*

$$\mathrm{KL}(p_X, q_X) < \epsilon ,$$

*and the base density be bounded. Let $\gamma^{-1} = \frac{\partial g}{\partial z} \frac{\partial g}{\partial z}^T$ be the inverse of the induced metric $\gamma$ in the canonical basis of coordinates $x$.*

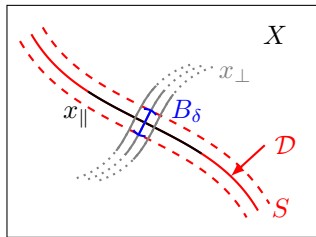

*Figure 2.* Gaussian normal coordinates, see Appendix D of (Carroll, 2019) for a detailed mathematical introduction.

*In this basis, $\gamma^{-1}$ is given by*

$$\gamma^{-1} = \begin{pmatrix} \gamma_{\mathcal{D}}^{-1} & & & \\ & \gamma_{B_{\delta_1}}^{-1} & & \\ & & \ddots & \\ & & & \gamma_{B_{\delta_{N-n}}}^{-1} \end{pmatrix} ,$$

*where $\gamma_{\mathcal{M}}^{-1}$ is the inverse of the induced metric on the submanifold $\mathcal{M} \in \{\mathcal{D}, B_{\delta_1}, \ldots, B_{\delta_{N-n}}\}$.*

*Furthermore, $\gamma_{B_{\delta_i}}^{-1} \to 0$ for vanishing radius $\delta_i \to 0$.*

We therefore conclude that gradient ascent in $Z$ space corresponds to gradient ascent in $X$ where the learning rate of all gradient components $\partial_{x_\perp} f$ orthogonal to the data manifold are effectively scaled by a vanishingly small learning rate. As a result, the gradient ascent in $Z$ will, to very good approximation, not leave the data manifold.

## 5. Experiments

**Tangent Space:** A non-trivial consequence of our theoretical insights is that we can infer the tangent plane of each point on the data manifold from our flow $g$. Specifically, we perform a singular value decomposition of the Jacobian $\frac{\partial g}{\partial z} = U \Sigma V$ and rewrite the inverse induced metric as

$$\gamma^{-1} = \frac{\partial g}{\partial z} \frac{\partial g}{\partial z}^T = U \Sigma^2 U^T . \quad (6)$$

For an approximately $n$-dimensional data manifold $S$ in an $N$-dimensional embedding space $X$, Theorem 2 shows that the inverse induced metric $\gamma^{-1}$ has $N - n$ small eigenvalues. The corresponding eigenvectors of the large eigenvalues will then approximately span the tangent space of the data manifold. In order to demonstrate this in a toy example, we train flows to approximate data manifolds with the shape of a helix and torus respectively. Figure 3 shows that we can indeed recover the tangent planes of these data manifolds to very good approximation. We refer to the Supplement B for details about the used flow and the data generation.

**Diffeomorphic Explanations:** We now demonstrate applications to image classification in several domains. The

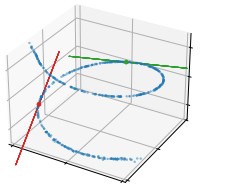 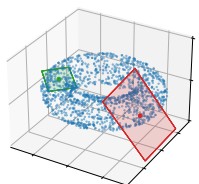

*Figure 3.* Approximate tangent planes for points on the data manifold $S$. As predicted by theory, the parallelepiped spanned by all *three* eigenvectors of the inverse induced metric scaled by the corresponding eigenvalues is to good approximation of the same dimension as the data manifold and tangential to it.

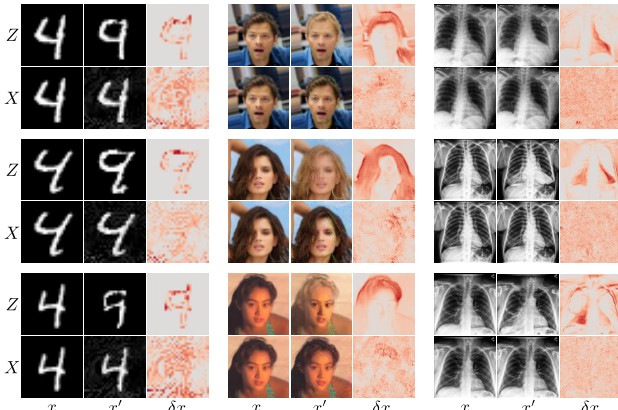

*Figure 4.* Counterfactuals for MNIST ('four' to 'nine'), CelebA ('non-blonde' to 'blonde'), and CheXpert ('healthy' to 'cardiomegaly'). Columns of each block show original image $x$, counterfactual $x'$, and difference $\delta x$ for three selected datapoints. First row is our method, i.e. gradient ascent in $Z$ space. Second row is standard gradient ascent in $X$ space. Heatmaps show sum over absolute values of color channels.

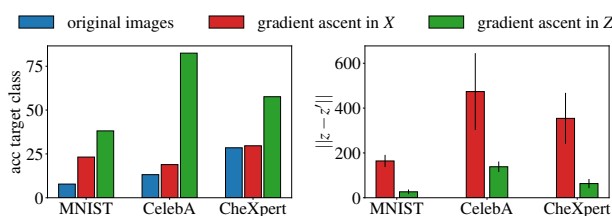

*Figure 5.* Generalization of counterfactuals to linear SVMs. Left: accuracy with respect to the target class $k$ generalizes better to SVM for $Z$-based counterfactuals. Right: distance in the base space is smaller for $Z$ than for $X$-based counterfactuals.

discussion is necessarily concise, see Supplement C for more details.

Datasets: We use the MNIST (Deng, 2012), CelebA (Liu et al., 2015), as well as the CheXpert datasets (Irvin et al., 2019). The latter is a dataset of labeled chest X-rays.

Classifiers: We train a ten-class classifier on MNIST (test accuracy of 99%). For CelebA, we train a binary classifier on the blonde attribute (test accuracy of 94%). For CheXpert, we train a binary classifier on the cardiomegaly attribute (test accuracy of 86%). All classifiers consists of a few standard convolutional, pooling and fully-connected layers with ReLU-activations and batch normalization.

Flows: We choose a flow with RealNVP-type couplings (Dinh et al., 2016) for MNIST and the Glow architecture (Kingma & Dhariwal, 2018) for CelebA and CheXpert.

Estimation of Counterfactuals: We select the classes 'nine', 'blonde', and 'cardiomegaly' as targets $k$ for MNIST, CelebA, and CheXpert, respectively, and take the confidence threshold to be $\delta = 0.99$. We use Adam for optimization.

Results: Counterfactuals produced by the flow indeed show semantically meaningful deformations in particular when compared to counterfactuals produced by gradient ascent in the data space $X$, see Figure 4. For Figure 5, we train a linear SVM for the same classification tasks and show that the flow's counterfactuals generalize better to such a simple model suggesting that they indeed use semantically more relevant deformations than conventional counterfactuals produced by gradient ascent in $X$ space.

## 6. Related Works

An influential reference for our work is (Singla et al., 2019) which uses generative adversarial networks (GANs) to generate counterfactuals, see also (Liu et al., 2015; Samangouei et al., 2018) for similar methods. Other approaches (Dhurandhar et al., 2018; Joshi et al., 2019) use Autoencoders instead of GANs. While both classes of generative models can currently sample more realistic high-dimensional samples, they are not bijective. As a result, an encoder network

has to be used which comes at the risk of mode-dropping and without any theoretical guarantees in contrast to our work. (Sixt et al., 2021) propose to train a linear classifier in the base space of a normalizing flow and show that this classifier tends to use highly interpretable features. In contrast to their approach, our method is completely model-agnostic. In (Rombach et al., 2020; Esser et al., 2020), an invertible neural network is used to decompose latent representations of an autoencoder into semantic factors to automatically detect interpretable concepts as well as invariances of classifiers.

## 7. Conclusion

In this work, we have used the fact that a normalizing flow is a diffeomorphism to map the data space to its base space. In this space, we can then straightforwardly perform gradient ascent on the data manifold, as we have established rigorously using Riemannian differential geometry. For future

work, more high-dimensional classification tasks will be considered as well as the dependence of the explanations on the chosen flow architecture. Furthermore, it would be interesting to evaluate the robustness of these explanations with respect to adversarial model and input manipulations (Ghorbani et al., 2019; Dombrowski et al., 2019; Anders et al., 2020; Heo et al., 2019).

## Acknowledgments

We want to thank the anonymous reviewers for their valuable and detailed feedback. A.K.D. is supported by the Research Training Group "Differential Equation- and Data-driven Models in Life Sciences and Fluid Dynamics (DAEDALUS)" (GRK 2433). J.G. is supported by the Swedish Research Council and by the Knut and Alice Wallenberg Foundation. P.K. is supported in part by the German Ministry for Education and Research (BMBF) under Grants 01IS14013A-E, 01GQ1115, 1GQ0850, 01IS18025A and 01IS18037A. P.K. also wants to thank Shinichi Nakajima for insightful discussions.

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

## A. Proofs

### A.1. Proof of Theorem 1

We repeat the theorem for convenience:

**Theorem.** *Let $z^{(t)}$ be defined as in (2) and $x^{(t)} = g(z^{(t)})$. Then, to leading order in the learning rate $\lambda$,*

$$x^{(t+1)} = x^{(t)} + \lambda\,\gamma^{-1}|_{g^{-1}(x^{(t)})}\,\frac{\partial f_k}{\partial x}(x^{(t)}) + \mathcal{O}(\lambda^2)\,, \quad (7)$$

*where $\gamma^{-1} = \frac{\partial g}{\partial z}\frac{\partial g}{\partial z}^T \in \mathbb{R}^{N,N}$ is the pull-back of the flat metric on $Z$ under the flow $g$.*

*Proof.* The step $x^{(t+1)} = g(z^{(t+1)})$ can be rewritten using the update formula (2) of the gradient ascent in $Z$ as

$$x^{(t+1)} = g\left(z^{(t)} + \lambda\frac{\partial(f_k \circ g)}{\partial z}(z^{(t)})\right)\,. \quad (8)$$

We now perform a Taylor expansion to leading order in the learning rate $\lambda$ using index notation as it eases notation

$$x_i^{(t+1)} = g(z^{(t)})_i + \lambda\sum_{j,l}\frac{\partial g_i}{\partial z_j}\frac{\partial g_l}{\partial z_j}\frac{\partial f_k}{\partial x_l}(g(z^{(t)})) + \mathcal{O}(\lambda^2)\,.$$

The result then follows by identifying $g(z^{(t)}) = x^{(t)}$ and $\gamma_{il}^{-1} = \sum_j \frac{\partial g_i}{\partial z_j}\frac{\partial g_l}{\partial z_j}$ which in matrix notation is given by $\gamma^{-1} = \frac{\partial g}{\partial z}\frac{\partial g}{\partial z}^T$. $\qquad\square$

### A.2. Proof of Theorem 2

Following the notation used throughout the main part, we denote by $p_X$ the data probability density. In particular, it holds that the data manifold is given by $S = \mathrm{supp}(p)$. The flow $g : Z \to X$ induces the probability density $q_X$ on the target space $X$ by push-forward of a base density $q_Z$ on the base space $Z$, i.e. $q_X(x) = q_Z(g^{-1}(x))|\frac{\partial z}{\partial x}|$.

Before giving the proof of Theorem 2, we will first derive the following result:

**Theorem 3.** *Let the flow be well-trained such that*

$$\mathrm{KL}(p_X, q_X) < \epsilon\,, \quad (9)$$

*for some small $\epsilon \in \mathbb{R}$. Then, we have for the data manifold $S \subset X$*

$$\int_S q_X(x)\,\mathrm{d}x > 1 - \epsilon\,. \quad (10)$$

*Proof.* By assumption,

$$-\mathrm{KL}(p, q) > -\epsilon\,.$$

Using the definition of the KL-divergence and the inequality $\ln(a) \le a - 1$, it the follows that

$$
\begin{aligned}
-\epsilon &< \int_S p_X(x)\,\ln\left(\frac{q_X(x)}{p_X(x)}\right)\,\mathrm{d}x \\
&\le \int_S p_X(x)\left(\frac{q_X(x)}{p_X(x)} - 1\right)\,\mathrm{d}x \\
&= \int_S q_X(x)\,\mathrm{d}x - 1\,,
\end{aligned}
$$

and thus

$$\int_S q_X(x)\,\mathrm{d}x > 1 - \epsilon\,. \quad (11)$$

$\qquad\square$

We repeat Theorem 2 for convenience:

**Theorem.** *Let $p_X$ denote the data density with $\mathrm{supp}(p_X) = S$, and the flow $g$ be well-trained such that*

$$\mathrm{KL}(p_X, q_X) < \epsilon\,,$$

*and the base density be bounded. Let $\gamma^{-1} = \frac{\partial g}{\partial z}\frac{\partial g}{\partial z}^T$ be the inverse of the induced metric $\gamma$ in the canonical basis of coordinates $x$.*

*In this basis, $\gamma^{-1}$ is given by*

$$\gamma^{-1} = \begin{pmatrix} \gamma_{\mathcal{D}}^{-1} & & & \\ & \gamma_{B_{\delta_1}}^{-1} & & \\ & & \ddots & \\ & & & \gamma_{B_{\delta_{N-n}}}^{-1} \end{pmatrix}\,,$$

*where $\gamma_{\mathcal{M}}^{-1}$ is the inverse of the induced metric on the submanifold $\mathcal{M}$.*

*Furthermore, $\gamma_{B_{\delta_i}}^{-1} \to 0$ for vanishing radius $\delta_i \to 0$.*

*Proof.* In the chosen coordinates, the metric $\gamma$ takes the block-diagonal form (in the canonical basis)

$$\gamma = \begin{pmatrix} \gamma_{\mathcal{D}} & & & \\ & \gamma_{B_{\delta_1}} & & \\ & & \ddots & \\ & & & \gamma_{B_{\delta_{N-n}}} \end{pmatrix},$$

see e.g. Example 13.2 of (Lee, 2013) for a proof. In these coordinates, we can then perform the integral (10) of Theorem 3:

$$1 - \epsilon < \int_S \left| \det \frac{\partial z}{\partial x} \right| q_Z(g^{-1}(x)) \, dx$$

$$= \int_S \sqrt{\det |\gamma|} \, q_Z(g^{-1}(x)) \, dx,$$

where in the second step, we have used the definition of the induced metric $\gamma = \frac{\partial z}{\partial x} \frac{\partial z}{\partial x}^T$ which implies that $\det |\gamma| = \det \left| \frac{\partial z}{\partial x} \right|^2$. Using the Gaussian normal coordinates, we can rewrite the integral as

$$\int_{\mathcal{D}} \sqrt{|\gamma_{\mathcal{D}}|} \prod_{i=1}^{N-n} \left( \int_{-\delta_i}^{\delta_i} \sqrt{|\gamma_{B_{\delta_i}}|} \, dx_\perp^i \right) q_Z(g^{-1}(x)) \, d^n x_\parallel.$$

Using the assumption that the base density $q_Z$ is bounded, i.e. $q_Z(z) \leq C$, we arrive at the inequality

$$1 - \epsilon < C \int_{\mathcal{D}} \sqrt{\det |\gamma_{\mathcal{D}}|} \prod_{i=1}^{N-n} \left( \int_{-\delta_i}^{\delta_i} \sqrt{|\gamma_{B_{\delta_i}}|} \, dx_\perp^i \right) d^n x_\parallel.$$
(12)

The integral however vanishes in the limit of vanishing radius $\delta_i$ since

$$\int_{-\delta_i}^{\delta_i} \sqrt{|\gamma_{B_{\delta_i}}|} \, dx_\perp^i \to 0 \qquad \text{for} \qquad \delta_i \to 0,$$

unless $\sqrt{|\gamma_{B_{\delta_i}}|} \to \infty$. Thus for the inequality (12) to hold the metric $\gamma_{B_{\delta_i}}$ has to diverge in the limit of vanishing $\delta_i$.

Since the induced metric $\gamma$ is block-diagonal, its inverse is given by

$$\gamma^{-1} = \begin{pmatrix} \gamma_{\mathcal{D}}^{-1} & & & \\ & \gamma_{B_{\delta_1}}^{-1} & & \\ & & \ddots & \\ & & & \gamma_{B_{\delta_{N-n}}}^{-1} \end{pmatrix}.$$

Because $\gamma_{B_{\delta_i}} \in \mathbb{R}$ diverges for vanishing radius, it follows that $\gamma_{B_{\delta_i}}^{-1} \to 0$ for $\delta_i \to 0$. $\square$

## B. Toy Example for Tangent Space

**Flow** The flow used for the toy example is composed of 12 RealNVP-type coupling layer blocks. Each of these blocks includes a three-layer fully-connected neural network with leaky ReLU activations for the scale and translation functions. For training, we sample from the target distribution. We train for 5000 epochs using a batch of 500 samples per epoch. We use the Adam optimizer with standard parameters and learning rate $\lambda = 1 \times 10^{-4}$. This takes around 10 minutes on a standard CPU.

**Latent distribution** We use a 3D standard Gaussian distribution as the latent distribution.

**Helix** To get a data sample from the helix we sample from a uniform distribution $x_3 \sim \mathcal{U}(-4, 4)$ and define $x_1 = \sin(x_3)$ and $x_2 = \cos(x_3)$.

**Torus** We define a torus with outer radius $R = 3$ and unit inner radius. To get a data sample from the Helix we sample from a uniform distribution $\phi, \theta \sim \mathcal{U}(0, 2\pi)$ and define $x_0 = \cos(\theta)(R + \cos(\phi))$, $x_1 = \sin(\theta)(R + \cos(\phi))$, and $x_3 = \sin(\phi)$.

## C. Details on Experiments

### C.1. Flows

**Architecture:** We use the RealNVP architecture[2] for MNIST and the Glow architecture[3] for CelebA and CheXpert.

**Training:** We use the Adam optimizer with a learning rate of $1 \times 10^{-4}$ and weight decay of $5 \times 10^{-4}$ for all flows. MNIST: we train for 30 epochs on all available training images. Bits per dimension on the test set average to 1.21. CelebA: we train for 8 epochs on all available training images. We use 5 bit images. Bits per dimension on the test set average to 1.32. CheXpert: we train for 4 epochs on all available training images. Bits per dimension on the test set average to 3.59.

### C.2. Classifier

**Architecture:** All classifiers have a similar structure consisting of convolutional, pooling and fully connected layers. We use ReLU activations and batch normalization. For MNIST we use four convolutional layers and three fully connected layers. For CelebA and CheXpert we use six convolutional layers and four fully connected layers.

---

[2]adapted from https://github.com/fmu2/realNVP
[3]adapted from https://github.com/rosinality/glow-pytorch

**Training:** We use the Adam optimizer with a weight decay of $5 \times 10^{-4}$ for all classifiers.

MNIST: we use training and test data as specified in torchvision. We use 10% of the training data for validation. We train for 4 epochs using a learning rate of $1 \times 10^{-3}$. We get a test accuracy of 0.99.

CelebA: we take training and test data set as specified in torchvision. We use 10% of the training images for validation. We scale and crop the images to 64×64 pixels. We partition the data sets into all images for which the blonde attribute is positive and the rest of the images. We treat the imbalance by undersampling the class with more examples. We train for 10 epochs using a learning rate of $5 \times 10^{-3}$. We get a balanced test accuracy of 93.63% by averaging over true positive rate (93.95%) and true negative rate (93.31%).

CheXpert: we choose the first 6500 patients from the training set for testing. The remaining patients are used for training. We select the model based on performance on the original validation set. We only consider frontal images and scale and crop the images to 128×128 pixels. For the training data the cardiomegaly attibute can have four different values: blanks, 0, 1, and -1. We label images with the blank attribute as 0 if the no finding attribute is 1, otherwise we ignore images with blank attributes. We also ignore images where the cardiomegaly attribute is labeled as uncertain. Using this technique, we obtain 25717 training images labelled as healthy and 20603 training images labelled as cardiomegaly. We do not treat the imbalance but train on the data as is. We train for 9 epochs using a learning rate of $1 \times 10^{-4}$. We test on the test set, that was produced in the same way as the training set. We get a balanced test accuracy of 86.07% by averaging over true positive rate (84.83%) and true negative rate (87.27%).

factuals and $\lambda = 5 \times 10^{-2}$ for counterfactuals found via the flow. We do a maximum of 2000 steps stopping early when we reach the target confidence of 0.99. We perform attacks on 500 images of the true class 'four'. All conventional attacks and 498 of the attacks via the flow reached the target confidence of 0.99 for the target class 'nine'.

For CelebA we use $\lambda = 7 \times 10^{-4}$ for conventional counterfactuals and $\lambda = 5 \times 10^{-3}$ for counterfactuals found via the flow. We do a maximum of 1000 steps stopping early when we reach the target confidence of 0.99. We perform attacks on 500 images of the true class 'non-blonde'. 492 conventional attacks and 496 of the attacks via the flow reached the target confidence of 0.99 for the target class 'blonde'.

For CheXpert we use $\lambda = 5 \times 10^{-4}$ for conventional counterfactuals and $\lambda = 5 \times 10^{-3}$ for counterfactuals found via the flow. We do a maximum of 1000 steps stopping early when we reach the target confidence of 0.99. We perform attacks on 1000 images of the true class 'healthy'. All conventional attacks and 990 of the attacks via the flow reached the target confidence of 0.99 for the target class 'cardiomegaly'.

## D. Examples for Counterfactuals

In this supplement, we present results on randomly selected images from the three datasets for which we produce counterfactuals via the flow. For the heatmaps, we visualize both the sum over the absolute values of color channels as well as the sum over the color channnels.

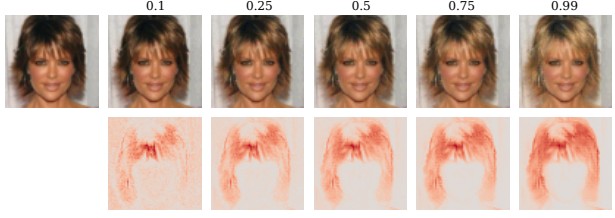

*Figure 6.* left: original image, first row: evolution throughout optimization. Numbers indicate confidence with which the image is classified as 'blonde'. Second row: heatmaps of $\delta x$

### C.3. Optimization Counterfactuals

Counterfactuals are found using the Adam optimizer with standard parameters. We vary only the learning rate $\lambda$.

For MNIST we use $\lambda = 5 \times 10^{-4}$ for conventional counter-

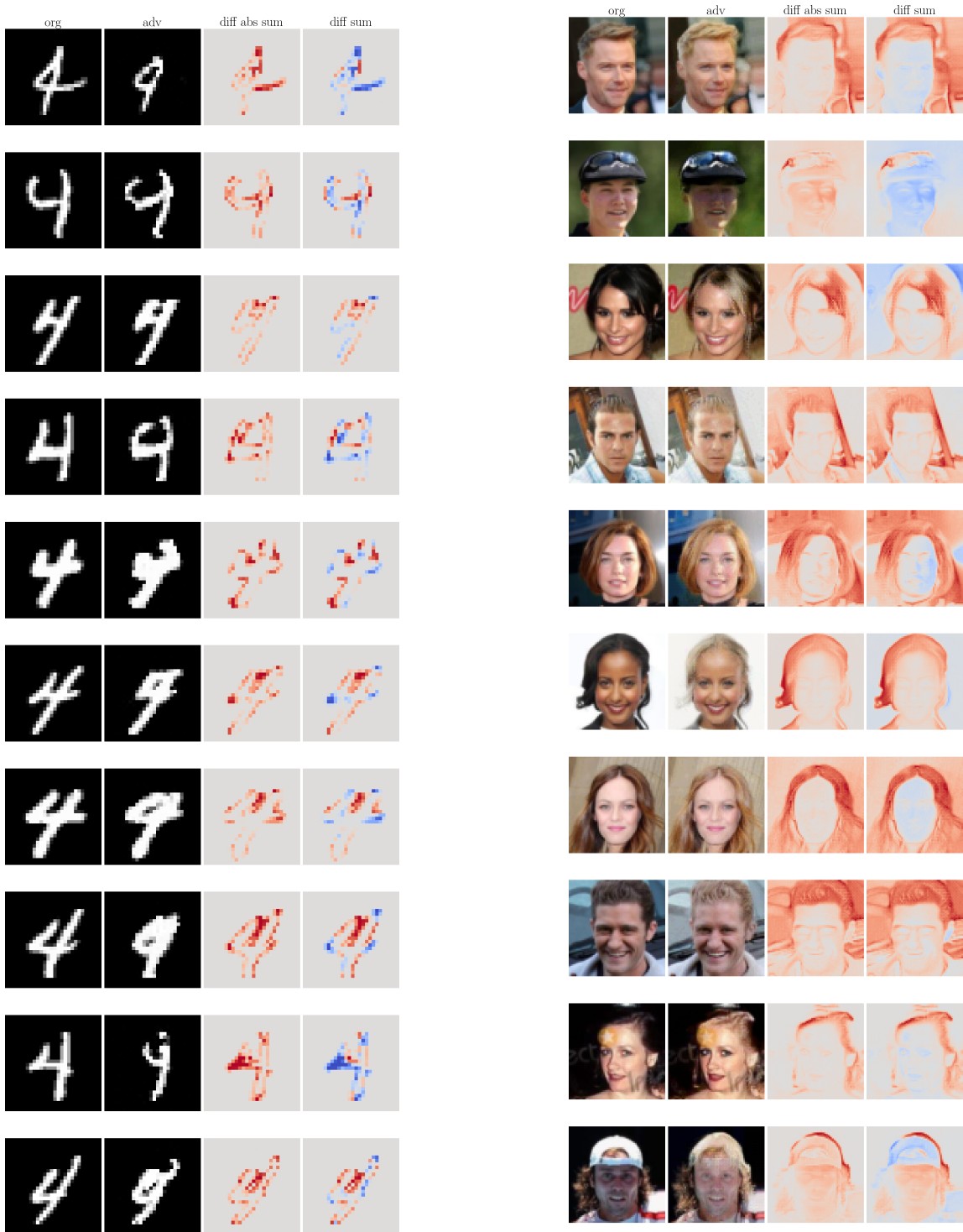

*Figure 7.* Randomly selected examples MNIST 'four' to 'nine'

*Figure 8.* Randomly selected examples CelebA 'not blonde' to 'blonde'

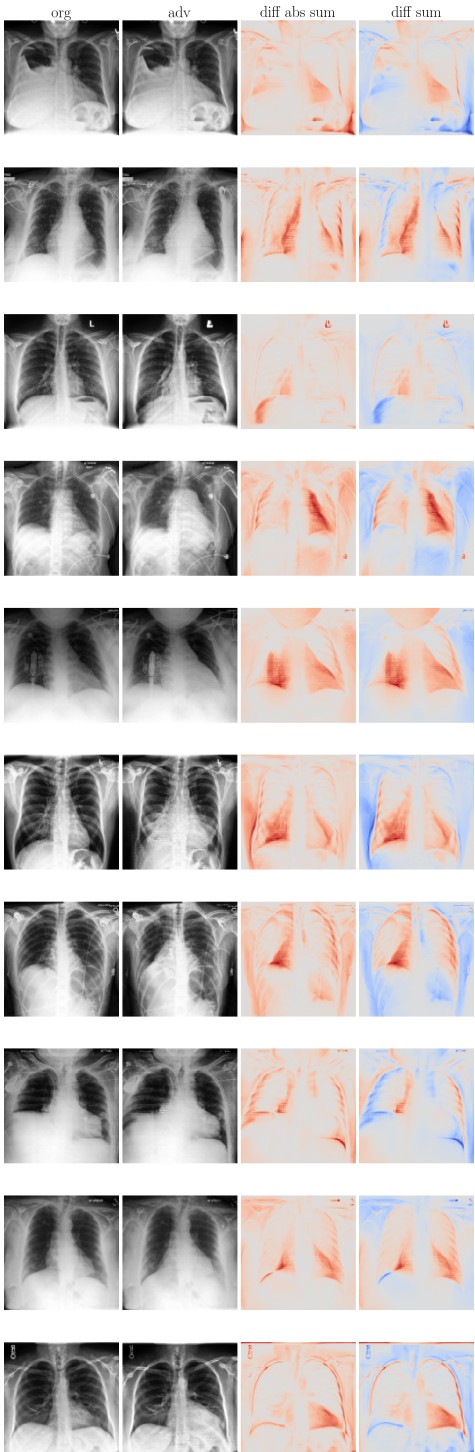

*Figure 9.* Randomly selected examples CheXpert 'healthy' to 'cardiomegaly'