# OpenReview forum: "Diffeomorphic Explanations with Normalizing Flows"
_ICML.cc/2021/Workshop/INNF — INNF+ 2021 contributedtalk_

### Official Review · Reviewer_bAQH · 2021-06-04

**Rating:** Borderline Accept
**Confidence:** 4

**Summary:**

The paper proposes to obtain the counterfactual of data w.r.t a classifier f by doing gradient descent in the latent space of a normalizing flow. Compared to other generative model-based approaches, the authors argue that the gradient descent on the latent space of normalizing flows ensures the resulting counterfactual lies in the data manifold, and the authors provide theoretical justifications on this.

**Justification For Rating:**

The idea of using normalizing flow for finding counterfactual is novel, although the overall approach is not very different from previous methods that use auto-encoder instead of NFs. Of course, being able to stay in the data manifold is good, but in practice, it is difficult to let NFs model the data manifold well. NFs usually have low sample quality when sampling z from un-truncated base distribution, suggesting that the data manifold is not well captured. In addition, since the hypothesis is that images lie on low dimensional manifold, does it make even more sense to use auto-encoders, where it is explicit to map data to a low dimensional manifold? Therefore, I think the paper needs more justification on the true advantage of NFs over AEs.

Nevertheless, the idea of running GD in latent space of NFs and mapping back to x-space is interesting and maybe useful for other tasks. The authors may want to cite some related papers in this direction:

NeuTra-lizing Bad Geometry in Hamiltonian Monte Carlo Using Neural Transport https://arxiv.org/abs/1903.03704

Exponential Tilting of Generative Models: Improving Sample Quality by Training and Sampling from Latent Energy https://arxiv.org/abs/2006.08100

Learning Energy-based Model with Flow-based Backbone by Neural Transport MCMC https://arxiv.org/abs/2006.06897

They all claim that it is beneficial to sample from the latent space of NFs by Langevin dynamics (which is essentially GD + noise).

---

### Official Review · Reviewer_YCDh · 2021-06-11

**Rating:** Accept
**Confidence:** 4

**Summary:**

This paper constructs counterfactual perturbations of images -- that is, perturbations which lie tangent to the data manifold, and hence are explainable. This is in contrast to adversarial perturbations, which fool a classifier but are not (in general) explainable.

The method is quite ingenious: perturbations are done in the "base" space, ie in the latent space (which is diffeomorphic to the data space), via a pullback using a normalizing flow. The authors show that gradient ascent in the latent space is (theoretically and empirically) sufficient to constrain perturbations to be (nearly) tangent to the data manifold. This is in contrast to ordinary gradient ascent in the data space (typically used in the adversarial robustness literature), which is not constrained to lie in the data manifold in any way.

**Justification For Rating:**

This is a good workshop paper, with an excellent idea. I feel that the idea is worthy of more than a brief 4-page workshop paper, and I would encourage the authors to submit a full-length version of this to a future conference venue.

Some things which could be improved:
1) for people without background in differential geometry, the language might be confusing. It would be helpful to explain differential geometric concepts intuitively in words (and with simple diagrams) for those readers. (Although admittedly it's not the authors job to do a crash course in DG)
2) I would have liked the authors spent more time emphasizing the merits of counterfactual perturbations over adversarial perturbations. This might be lost on some readers -- counterfactuals really are a useful framework, and bears emphasizing, especially since DL has been obsessed with adversarial perturbations for so long.

---

### Official Review · Reviewer_tbxe · 2021-06-12

**Rating:** Accept
**Confidence:** 4

**Summary:**

The paper proposes a method to generate counter-factual examples by performing gradient ascent in the base space of a normalizing flow. The work justifies the proposed method with theoretical analysis showing the obtained counter-factual example will stay close the data manifold. The theoretical analysis conclusion is further backed by experiments of recovering tangent space using synthetic data. The validity of the proposed method to generate counter-factual examples is supported by quantitative study and qualitative samples on real datasets.

**Justification For Rating:**

The method of generating counter-factual examples using a normalizing flow learning the data distribution is novel itself. The study and justification the proposed method are also thorough and solid. The work justifies the method with proper mathematical assumption of data manifold and rigourous theoretical analysis. The conclusion of the theoretical analysis provides important insights into why the generated counter-factual example stay close to the data manifold and the advantages of the proposed method over gradient ascent in data space. The work also designs a convincing experiment on synthetic data to back the conclusion of theoretical analysis. Both quantitative and qualitative experiment results on image data support the arguments and contribution of the work. The quality of organization and presentation is satisfying. There’re still some typos in the paper. For example, in line 133 to 135 and line 140, it is better to replace Equation (7) which is in the appendix with Equation (4). In line 275, Theorem 1 should be Theorem 2.

---

### Decision · Program_Chairs · 2021-06-14

Accept (contributed talk)